# Identification and Characterization of Aptamers Targeting Ovarian Cancer Biomarker Human Epididymis Protein 4 for the Application in Urine

**DOI:** 10.3390/cancers15020452

**Published:** 2023-01-10

**Authors:** Antonija Hanžek, Frédéric Ducongé, Christian Siatka, Anne-Cécile E. Duc

**Affiliations:** 1UPR CHROME—Université de Nîmes, CEDEX 1, 30021 Nîmes, France; 2CEA, Fundamental Research Division (DRF), Institut of Biology François Jacob (Jacob), Molecular Imaging Research Center, 92260 Fontenay-aux-Roses, France

**Keywords:** aptamers, cancer, ovarian cancer, HE4, SELEX, ddPCR, sequencing, diagnostics, urine

## Abstract

**Simple Summary:**

Ovarian cancer is the deadliest gynecological cancer. Due to the lack of effective diagnostic methods and the non-specific symptoms of the disease, late diagnosis remains a main factor of the poor prognosis. Therefore, development of novel diagnostic approaches are needed. Recently, urine has become an interesting non-invasive source of cancer biomarkers. Human epididymis protein 4 (HE4) is a protein overexpressed in ovarian cancer, but not in healthy or benign conditions. In urine, HE4 stands as a biomarker with high stability and diagnostic value for detection of ovarian cancers. Recently, aptamers emerged as inexpensive detection probes for cancer detection. Aptamers are single-stranded oligonucleotides that bind with high affinity to target molecules. Here, we selected, identified and characterized DNA aptamers targeting human HE4 in urine, with the affinities in the nanomolar range. Therefore, they could represent a promising tool for application in diagnostics and future development of urine tests or biosensors for ovarian cancer.

**Abstract:**

Ovarian cancer is the deadliest gynecological cancer. With non-specific symptoms of the disease and the lack of effective diagnostic methods, late diagnosis remains the crucial hurdle of the poor prognosis. Therefore, development of novel diagnostic approaches are needed. The purpose of this study is to develop DNA-based aptamers as potential diagnostic probes to detect ovarian cancer biomarker Human epididymis protein 4 (HE4) in urine. HE4 is a protein overexpressed in ovarian cancer, but not in healthy or benign conditions. With high stability and diagnostic value for detection of ovarian cancer, urine HE4 appears as an attractive non-invasive biomarker. The high-affinity anti-HE4 DNA aptamers were selected through 10 cycles of High Fidelity Systematic Evolution of Ligands by EXponential enrichment (Hi-Fi SELEX), a method for aptamer selection based on digital droplet PCR. The anti-HE4 aptamers were identified using DNA sequencing and bioinformatics analysis. The candidate aptamer probes were characterized in urine for binding to HE4 protein using thermofluorimetry. Two anti-HE4 aptamers, AHE1 and AHE3, displayed binding to HE4 protein in urine, with a constant of dissociation in the nanomolar range, with K_d_ (AHE1) = 87 ± 9 nM and K_d_ (AHE3) aptamer of 127 ± 28 nM. Therefore, these aptamers could be promising tools for application in diagnostics and future development of urine tests or biosensors for ovarian cancer.

## 1. Introduction

Ovarian cancer (OC) is the deadliest gynecological cancer [1]. It is the 8th most common cancer type in the world and the 9th cause of all cancer-related deaths [2]. With the global aging of the population, OC may become a public health problem, with the prediction of a 55% increase in the global incidence by 2035 [3]. Only about 20% of ovarian cancers are found at an early stage. When ovarian cancer is found early, about 94% of patients live longer than 5 years after diagnosis [1,2]. Unfortunately, most of the cases are diagnosed at advanced stages of disease, with a five-year survival rate dropping to 40 to 15% [4,5]. Due to the non-specific symptoms and lack of efficient diagnostic methods, late diagnosis remains the main contributing factor to the high mortality [6]. Conventional diagnostic methods include pelvic examination, transvaginal ultrasound and blood test of Cancer Antigen 125 (CA125), but they are not sensitive, nor specific enough to detect the disease in its early stages [5,6,7]. Another important biomarker is a Human epididymis protein 4 (HE4), a glycoprotein overexpressed in ovarian cancer, but not in benign conditions or healthy individuals [8]. HE4 is a secreted protein, found in body fluids, including blood, urine and ascites of patients [9,10]. Therefore, serum HE4 is used as a clinical biomarker in management of OC, for the monitoring of the therapy’s efficiency and detection of the recurrence [11]. As a single biomarker, it has the highest specificity and sensitivity for OC detection [11,12]. Serum HE4 was approved by the Food and Drug Administration in 2007 for differential diagnosis of women with pelvic masses [13].

Compared to CA125, HE4 has several advantages, as HE4 serum levels are not elevated during pregnancy, menstruation or in benign gynecological conditions [6]. Moreover, it is elevated earlier in the course of the disease [14]. A risk of malignancy algorithm (ROMA) is a more recent test, which combines the serum CA125 and HE4 with menopausal status into a numerical score that predicts the risk for cancer [13]. The FDA has approved ROMA for distinguishing malignant from benign pelvic masses in 2011 and is used globally today, as it significantly increases diagnostic sensitivity and specificity for distinguishing between benign and malignant disease [13]. Measuring HE4/ROMA is more reliable for diagnosing ovarian cancer than CA125 alone and it is implemented as a standard test in conjunction with gynecological examinations in routine clinical practice.

Gynecological examinations and blood tests can sometimes be considered invasive or intrusive, often delaying the time in which women take in deciding to make a doctor’s appointment [15]. In addition, there is no screening method of the general population available for OC [16]. Therefore, more effective and less invasive diagnostic approaches are needed in order to improve the poor prognosis. Recently, urine has become an interesting source of OC biomarkers that could provide an easy, cheap and non-invasive alternative in cancer diagnostics [17,18,19]. Due to the small size of 25 kDa, which is below the limit of glomerular filtration, it makes HE4 an ideal urine target biomarker of ovarian cancer. In 2010, Hellstrom and colleagues first identified HE4 in the urine of OC patients with a high specificity for the detection of cancer [20]. Since then, the diagnostic value of urine HE4 has been investigated [21,22,23]. It has been shown that urine HE4 has similar diagnostic value as in serum, with a specificity of 92% and a sensitivity of 76%, respectively [24]. Therefore, urine represents an easier, non-invasive source of HE4. Moreover, urine HE4 is more stable for longer periods of time compared to serum HE4 or CA125, especially in patients with chemotherapy resistance [22]. It is also an efficient tool in the detection of recurring OC, as urine HE4 is positive earlier than serum HE4 or CA125, prior to the clinical diagnosis [22]. Additionally, urine HE4 is present in higher concentration than in serum, which makes it an ideal target for future development of the diagnostic test for OC.

Recently, aptamers have emerged as interesting diagnostic tools for cancer detection [25]. Aptamers are short single-stranded DNA or RNA oligonucleotides that bind with high affinity and specificity to a wide range of targets, including small molecules, proteins or even whole cells [26]. Their intramolecular forces enable formation of three-dimensional structures which allow specific recognition and binding of the target. Aptamers are obtained by in vitro selection, using various formats of Systematic Evolution of Ligands by EXponential enrichment (SELEX) [27]. Aptamers have a myriad of applications in the cancer research field, ranging from use as tools in clinical diagnosis, to drug delivery and cancer therapeutic agents [28]. In diagnostics, they are gaining popularity due to advantages of conventional antibodies. Compared with antibodies, aptamers also possess characteristics of high affinity and specificity to bind to oncological markers, while having numerous advantages in a variety of the chemical modifications, stability and production cost. Therefore, they have been used in different fields of tumor diagnosis, such as in detection of the tumor biomarkers or cancer cells, immunohistochemical analysis, and in vivo imaging [28].

In the recent years, only several aptamers have been developed to target OC protein biomarkers. The majority of the aptamers have been developed to target CA125 [29]. Until now, no aptamer for OC has been previously developed or applied in urine. Herein, we present selection, identification and characterization of high-affinity DNA aptamer probes targeting HE4 for the application in urine, for potential future urine tests or biosensors for OC.

## 2. Materials and Methods

### 2.1. Aptamers and Proteins

The starting aptamer pool was a 70 nucleotide long (70 nt) single-stranded DNA library (Integrated DNA Technologies, Inc., Coralville, IA, USA) with 30 nucleotide long (30 N) random region flanked by two fixed, 20 nucleotide long (20 nt) constant primer binding sites: TCGCACATTCCGCTTCTACC–N_30_–CGTAAGTCCGTGTGTGCGAA, containing ~ 4^25^ = 1^15^ highly diverse random sequences. The library was ordered on a scale of 10 µmol DNA and resuspended in nuclease-free H_2_O. To ensure equal nucleotide distribution, the molar ratio of A:C:G:T phosphoramidites was optimized to a ratio 29:29:20:22%. The primers (Eurofins Genomics, Ebersberg, Germany) for aptamer amplification have been selected from the literature and previously utilized in SELEX: Forward 5′-TCG CAC ATT CCG CTT CTA CC-3′ and Reverse 5′-[phos]-TTC GCA CAC ACG GAC TTA CG-3′ [30]. The reverse primer is 5′phosphorylated for the ssDNA regeneration by the lambda exonuclease.

70-mer Anti-HE4 aptamers (Eurofins Genomics, Ebersberg, Germany): AHE1: TCG CAC ATT CCG CTT CTA CCC CCA ACC AAC GTC TAT ACT TCC CCA ACC TCC GTA AGT CCG TGT GTG CGA A; AHE2: TCG CAC ATT CCG CTT CTA CCG CCA ACA TCG TAC TCC ATC TGC CAC CCC CAC GTA AGT CCG TGT GTG CGA A; AHE3: TCG CAC ATT CCG CTT CTA CCC ATC AAT CAG CAT ACC CAC CAT GTA CGT CCC GTA AGT CCG TGT GTG CGA A. Scramble aptamer DNA (negative control): ATC GCG TCC GTC AAT TAC CTA CGC TGC ACC ACT TAT GAG CCG GTA TCG CCA TCC GTC AAC TCA CGC CTC C.

Human epididymis protein 4 is a 6xhistidine-tagged human recombinant protein, expressed in HEK293 cells (ref. ab219658, Abcam, Cambridge, UK). Counter-selection was performed to 6xhistidine peptide (ref. RP11737, Genscript Biotech Corp., Piscataway, NJ, USA).

### 2.2. Urine Preparation

Following, 1× urine was prepared to correspond to human urine [31], and it contains 11.965 mM Na_2_SO_4_, 1.487 mM C_5_H_4_N_4_O_3_, 2.450 mM Na_3_C_6_H_5_O_7_·2H_2_O, 7.791 mM C_4_H_7_N_3_O, 249.750 mM CH_4_N_2_O, 30.953 mM KCl, 30.053 mM NaCl, 1.663 mM CaCl_2_, 23.667 mM NH_4_Cl, 0.19 mM K_2_C_2_O_4_·H_2_O, 4.389 mM MgSO_4_·7H_2_O, 18.667 mM NaH_2_PO_4_·2H_2_O, 4.667 mM Na_2_HPO_4_·2H_2_O. The urine is prepared a day before for pH stabilization (measured pH = 6.3). All the reagents were supplied from Sigma-Aldrich (Sigma-Aldrich, St. Louis, MO, USA).

### 2.3. HE4 Protein Immobilization

The HE4 protein was immobilized on HisPur™ Ni-NTA beads (ref. 88223, Thermo Fisher Scientific, Waltham, MA, USA). The beads were centrifuged at 700× *g* at room temperature (RT) for 1 min to remove the storage liquid. Then, the beads were washed with a two-volume of equilibrium buffer (20 mM sodium phosphate, 300 mM sodium chloride with 10 mM imidazole; pH 7.4) and centrifuged again at 700× *g* at RT for 1 min. The prepared HE4 protein sample was added and incubated at RT with shaking at 1500 rpm for 30 min to immobilize the protein. Depending on the cycle of SELEX, a protein sample was prepared to correspond to 100 pmol or 200 pmol HE4 on solid beads. After incubation, the beads were centrifuged at 700× *g* at RT for 1 min and solid beads with immobilized HE4 were subjected to DNA in SELEX.

### 2.4. Digital Droplet PCR-Based High-Fidelity SELEX to Ovarian Cancer Biomarker HE4

The anti-HE4 aptamers were selected after 10 cycles of selection using a modified version of digital droplet PCR (ddPCR)-based High Fidelity Systematic Evolution of Ligands by EXponential enrichment (Hi-Fi) SELEX (Figure 1). The selection conditions and stringency, such as protein and DNA amount and variations in washing steps were modified from round to round to ensure a high amount of recovery and high affinity (see details in Table 1 and Appendix A). After 8 cycles of selection to 6xhistidine-HE4, the recovered DNA aptamers were separated into two equal branches and subjected to an additional 2 rounds of positive selection (6xhistidine-HE4 on beads) and 2 rounds of counter-selection (6xhistidine peptide on beads), to identify potential non-specific binders to the sample matrix (protein tag and charged beads). Before each cycle, DNA was denatured for 5 min at 95 °C then placed on ice for 10 min, cooled down to RT and introduced to urine to ensure proper folding. Then, 1.25 nmol of the highly diverse DNA aptamer library was incubated with 200 pmol of target human HE4 protein immobilized on Ni-NTA beads for 1 h at 25 °C with shaking at 1500 rpm in the volume of 200 μL of 1× urine. The unbound sequences were eliminated by 3 steps of washing and increased stringency of washing (Appendix A). The specific DNA was dissociated from aptamer-HE4 protein complex by heating for 10 min at 95 °C, followed by DNA recovery using Phenol:Chloroform:Isoamyl alcohol (25:24:1 *v*/*v*) extraction and ethanol/GlycoBlue™ (ref. AM9515, Thermo Fisher Scientific) co-precipitation.

Then, the purified DNA was prepared for droplet digital PCR, in duplicates for (1) the droplet analysis—monitoring and quantification of the anti-HE4 sequences during SELEX and (2) the extraction from droplets after ddPCR to proceed to the next round of SELEX with the amplified DNA. The droplet partitioning allows for the rare anti-HE4 aptamer sequences amplification within the droplets. A ddPCR reaction includes 2 μL of SELEX selected aptamer template (unknown concentration), 10 μL 1× QX200™ ddPCR™ EvaGreen Supermix (ref. 1864034, Bio-Rad Laboratories, Inc., Hercules, CA, USA), 0.2 µL of 10 µM stock forward primer to a final 100 nM concentration in PCR reaction, 0.2 µL of stock 10 µM phosphorylated reverse primer to a final 100 nM concentration in PCR reaction and nuclease-free water to final reaction volume of 20 µL. Positive ddPCR using 2 µL of initial library (<10^−7^ ng DNA) as a template and a negative non-template control (NTC) using 2 µL of water were systematically included. The details of the ddPCR reactions and volumes are available in the Appendix A. Exactly 20 µL of PCR mixture and 70 μL of QX200™ Droplet Generation Oil for EvaGreen per reaction (ref. 1864006, Bio-Rad Laboratories, Inc., Hercules, CA, USA) were transferred into ddPCR DG8™ Cartridges (ref. 1864008, Bio-Rad Laboratories, Inc., Hercules, CA, USA) for the droplet generation in a QX200™ Droplet Generator (Bio-Rad Laboratories, Inc., Hercules, CA, USA). Then, generated ddPCR droplets were transferred into ddPCR plates (ref. 17005224, Bio-Rad Laboratories, Inc., Hercules, CA, USA) and heat sealed at 180 °C for 5 s using Bio-Rad PX1 PCR Plate Sealer (Bio-Rad Laboratories, Inc., Hercules, CA, USA). The ddPCR was performed for 40 cycles following an optimized EvaGreen program: 5 min at 95 °C (enzyme activation), 30 s at 95 °C (denaturation), 1 min at 60 °C (annealing/extension), 5 min at 4 °C (signal stabilization) and 5 min at 90 °C (signal stabilization) on a C1000 Touch Thermal Cycler (Bio-Rad Laboratories, Inc., Hercules, CA, USA). The amplified droplets were analyzed on QX200™ ddPCR System (Bio-Rad Laboratories, Inc., Hercules, CA, USA) using QuantaSoft™ Software (Bio-Rad Laboratories, Inc., Hercules, CA, USA).

Amplified DNA was recovered from the droplets floating on the top of the wells immediately after ddPCR. Then, chloroform extraction was performed by adding 20 μL of Tris-HCl buffer (pH 7.4) and 70 μL of chloroform per well, followed by vortexing and centrifugation at 15,500× *g* for 10 min at RT. The upper aqueous phase, containing the recovered DNA, was transferred to a fresh tube. To increase the yield, DNA was re-amplified by multiple PCR reactions using a 2 µL DNA template, 4 μL of 10 μM stock forward primer to a final 400 nM concentration in the PCR reaction, 4 μL of 10 μM stock phosphorylated reverse primer to a final 400 nM concentration in PCR reaction, 0.5 μL of 5 U/µL DreamTaq polymerase (ref. EP0705, Thermo Fisher Scientific, Waltham, MA, USA), 10 µL of 10× DreamTaq polymerase buffer containing 20 mM MgCl_2_ (ref. EP0705, Thermo Fisher Scientific, Waltham, MA, USA) to a final concentration of 1× including 2 mM MgCl_2_, 2 µL of 10 µM stock dNTPs (ref. R0191, Thermo Fisher Scientific, Waltham, MA, USA) to a final concentration of 0.2 µM each in PCR reaction and nuclease-free H_2_O to a final of 100 µL reaction. The details of the ddPCR reactions and volumes are available in the Appendix A. The DNA is amplified on C1000 Touch Thermal Cycler (Bio-Rad Laboratories, Inc., Hercules, CA, USA) following PCR program: 95 °C for 5 min (enzyme activation), 95 °C for 30 s (denaturation) and 60 °C for 1 min (annealing/extension) for 10–30 cycles depending on the SELEX cycle, to obtain a single amplicon without by-products (Appendix A).

The pooled PCR reactions (total 30 reactions = 3 mL) were concentrated using Vivacon^®^ MWCO 10 kDa (Sartorius, Goettingen, Germany) with centrifugation at 4000× *g* for 20 min at 4 °C. Primers were removed using MicroSpin G-50 (Cytiva, Marlborough, MA, USA) columns with buffer exchange to lambda exonuclease buffer (67 mM glycine-KOH, 2.5 mM MgCl_2_).

The eluted double-stranded DNA was subjected to optimized ssDNA regeneration using 10 U/µL lambda exonuclease (ref. EN0561, Thermo Fisher Scientific, Waltham, MA, USA) for 1 h at 37 °C followed by enzyme deactivation for 10 min at 80 °C. Samples were pooled and the final ssDNA was purified from the enzyme by Phenol:Chloroform:Isoamyl alcohol (25:24:1 *v*/*v*) and ethanol precipitation.

The purified aptamer ssDNA was quantified by NanoDrop™ 2000c spectrophotometry (Thermo Fisher Scientific, Waltham, MA, USA) and validated by 4% agarose gel electrophoresis before using it as input for the next round of Hi-Fi SELEX. The amount of the DNA used in each cycle is calculated for our specific, small and single-stranded DNA sequence. The concentrations of selected aptamers at the end of each cycle (mol/L) were calculated based on Beer–Lambert law following the formulas: c (aptamers) = A_260_/(ε × l), where absorbance measurements A_260_ were obtained by NanoDrop™ 2000c spectrophotometry on 1 cm path with molar extinction coefficient ε of 654,479 L/(mol × cm). The concentration of selected aptamers at the end of each cycle (ng/µL) were calculated by formula c (ng/µL) = c (mol/L) × MW × 1000 where MW is molecular weight of the initial native 70 bases aptamer library (21,452.5 g/mol). Then, the corresponding amount of DNA was added to each cycle of selection corresponding to amount as shown in Table 1. The amount of DNA was depending on the amount of DNA recovered during the experimental procedure. The amount of DNA from cycle 6 to the end of selection was increased to increase the stringency of selection to target HE4.

### 2.5. DNA Sequencing and Bioinformatics

To study the enrichment of the anti-HE4 aptamers to target HE4 in urine, aliquots of the library after several rounds of SELEX were prepared for deep sequencing on the Illumina system as previously described [32]. Approximately 200,000 sequencing reads were analyzed for the starting library and for each round of SELEX from round 5 using a homemade software PATTERNITYseq (access to this software can be found at MIRCEN Plateforms, CEA, France at https://jacob.cea.fr/drf/ifrancoisjacob/english/Pages/Departments/MIRCen/Platforms/Aptamers.aspx, accessed on 4 January 2023). This analysis has been previously described [33]. Basically, the adapter and primer sequences were first removed from each sequence, leaving only the variable regions with a size between 25 and 32 nucleotides. Then, the frequency of each sequence in the different libraries was calculated and any sequences with a frequency <0.01% in all libraries were removed to decrease the time of analysis. The remaining sequences (2621 in our case) were then sequentially clustered in families using a Levenshtein distance of 6 (i.e., sequences with no more than 6 substitutions, insertions or deletions). Finally, the frequency of each family (2189 in our case) was calculated at every cycle. A partially conserved motif between the 10 most enriched families was searched using the MEME suite (https://meme-suite.org/meme/, accessed on 3 January 2023) [34]. The secondary structures of anti-HE4 aptamers were predicted using online DNA folding software Unafold (http://www.unafold.org/, accessed on 30 August 2022) [35], using urine salt concentration of [Na^+^] = 55.4 mM and [Mg^2+^] = 4.4 mM at temperature 25 °C.

### 2.6. Aptamers-HE4 Protein Binding Analysis in Urine by Thermofluorimetry

The bindings of the anti-HE4 aptamers to HE4 protein were analyzed using Thermofluorimetric analysis (TFA) found in literature [36,37]. DNA was denatured in an aptamer buffer (10 mM Tris–HCl, 50 mM KCl, 3.3 mM MgCl_2_; pH = 8.0) for 5 min at 95 °C, placed 10 min on ice before cooling down to RT. DNA was then diluted in urine and incubated with different concentrations of HE4 for 1 h at 25 °C. The constant aptamer concentration was subjected to the HE4 protein in concentrations with a range from 0–800 nM in 1/125 diluted urine. The reaction mixture consisted of 100 nM DNA in 1/125 urine, 1× Sybr-Gold dye (ref. S11494, Invitrogen, Waltham, MA, United States) and HE4 in protein buffer (10.1 mM Na_2_HPO_4_, 137 mM NaCl, 2.7 mM KCl, 1.8 mM KH_2_PO_4_, 1 mM MgCl_2_) to a final concentration ranging from 0–800 nM. The melting profile of aptamer-HE4 binding was analyzed on C1000 Touch/CFX96 Deep Well real-time system (Bio-Rad Laboratories, Inc., Hercules, CA, USA) with a temperature gradient ranging from 4 °C to 90 °C at 0.5 °C/min. Fluorescence was measured with excitation at 450/490 nm and detection at 515/530 nm. The melting profile was constructed by plotting the negative derivative fluorescence signal −d(RFU)/dT against HE4 concentration. New T_m_, corresponding to the melting of the aptamer-HE4 complex was observed after blank subtraction (blank was signal from free aptamer only, without protein). Finally, the binding curve was constructed from the negative derivative fluorescence signal −d(RFU)/dT at the T_m_ of the bound aptamer and concentration of the HE4 protein. The estimation of the constant of dissociation (K_d_) was performed using the online platform https://mycurvefit.com on the nonlinear sigmoidal regression model.

## 3. Results

### 3.1. Selection and ddPCR Amplification of Anti-HE4 Diagnostic Aptamers

A modified variant of the ddPCR-based, Hi-Fi SELEX method was used to select DNA aptamers specifically binding to the ovarian cancer biomarker HE4 (Figure 1). This method is based on digital droplet polymerase-chain reaction (ddPCR) amplification to ensure non-biased amplification. This approach enables partitioning of the individual DNA pools into droplets for homogenous sensitive amplification. The goal is to keep and amplify potential high-affinity rare sequences, while simultaneously enabling absolute quantification and monitoring the sequence pools. As described, 10 cycles of positive selection steps were performed to target human HE4 in urine. The conditions obtained at each SELEX round are reported in Table 1. At each cycle, aptamer DNA was validated by gel electrophoresis, to ensure proper 70 bp amplification, as well as complete conversion to 70 bases ssDNA using lambda exonuclease (Appendix A).

In ddPCR, droplets are classified into positive (anti-HE4 target DNA aptamers) and negative (background DNA) clusters based on the fluorescence threshold. The droplet is considered positive if there is at least 1 copy of the anti-HE4 aptamer sequence present. Many aptamer sequences were detected and amplified in all 10 cycles of SELEX (Figure 2A). The early rounds were performed using soft washing conditions, and as expected, high amounts of DNA were found in the early rounds, such that we observed saturation of droplets due to the excess of 70-mer ssDNA template (fluorescence ~ 5000–6000 RFU). Therefore, the amplification at those stages may not have been optimal, as we observed a very high number of sequences present (maximum sequences detected (>1 × 10^6^) at the very beginning of selection (C1+ to C4+). From cycle 5 (C5+), the amount of DNA was reduced and sequence diversity was reduced, while specificity for HE4 increased. Then optimal classification into positive and negative clusters with 70 bp dsDNA amplicons were observed (fluorescence ~ 10,000–12,000 RFU), presumably due to enrichment to target HE4. Positive ddPCR control (library) is important to ensure the amplification is working and to use it as a reference to track expected height of the specific amplicon fluorescence. Negative PCR control (water) should not have any amplification and all droplets should be classified as negative (Figure 2A).

The droplets can be qualified as total, positive and negative and then counted. The accurate absolute quantification of the aptamer sequences is achieved if the total number of droplets is >10,000, so total droplet count is used as a quality assurance tool. As the selection of aptamer against HE4 moves forward, a clearer repartition into positive and negative droplets was observed after cycle 5 (C5+). As selection progressed, the number of positive droplets decreased, while negative droplet number increased, probably due to the lower amount of DNA template and with potentially more specific sequences present amplified (Figure 2B). After C5+, it seemed that many sequences were successfully amplified and detected as bound to HE4, although the number varied in each cycle throughout the selection process (Figure 2C).

The last two positive selection steps (C9+ and C10+) to HE4 and two rounds of counter selection (C9− and C10−) to 6xhistidine peptide immobilized on Ni-NTA beads (no HE4) were performed, after equally distributing previously recovered DNA (C8+) to each branch of selection. This step is essential to identify the high-affinity sequences to ovarian cancer HE4 target, and to eliminate non-specific binders with affinity to sample matrix (no target). As the DNA is negatively charged and Ni-NTA beads are positively charged, it is expected that some aptamers could exhibit binding properties due to electrostatic interactions. Indeed, DNA was found to be amplified in both positive and counter selection. However, most importantly, a significantly higher number of aptamer sequences were present in positive selection to HE4 protein, compared to that in counter selection, suggesting high affinity and potential enrichment to target HE4 protein (Figure 2C). These conclusions were validated by deep sequencing, with specific sequences only being enriched to target HE4 and not to peptide tag or beads (Figure 3 and Figure 4).

### 3.2. DNA Sequencing and Identification of the Enriched Aptamers to HE4

To study the enrichment through the Hi-Fi selection procedure, deep sequencing was performed to analyze the starting library and the libraries after cycle 5 (Figure 3). In order to focus our analysis on the most enriched sequences, we have retained for analysis only 2621 sequences whose frequency was greater than 0.01% in at least one cycle. As expected, the starting library contained a large diversity of sequences where each sequence was at a very low frequency. In contrast, the library contained sequences whose frequency had strongly increased from the fifth cycle (Figure 3A,B). Among these sequences, some have been regrouped into a common family when they were separated by an edit distance lower than 6. This clustering created 2,189 families, the most abundant of which contained 29 sequences. It is interesting to note that the number of sequences and families increased between cycle 5 and 6 and then decreased during the following cycles (Figure 3B,C) but their percentage in the bank remained constant despite the increase of the selection pressure except for the last cycle (Figure 3A). This suggests an evolutionary pattern where some sequences and families increase in the library while others less adapted decrease. In addition, the number of sequences and families decreased significantly more in the last two cycles in the absence of protein and, especially, their percentage in the library was about 50% less (Figure 3A,E), suggesting that many sequences should bind to the HE4 protein. This was the case for the 10 most enriched families whose frequency increased to represent more than 1% of the library for some of them in the two last cycles with the protein, while it was about 5 to 10 times less without it (Figure 3D). A multiple alignment of the most amplified sequences did not reveal the presence of a conserved motif (Appendix A), however, the MEME suite identified a partially conserved motif of 7 nucleotides in the 10 most enriched anti-HE4 aptamer families (Appendix A). Therefore, we chose to evaluate the affinity of the 3 most enriched families in the last cycles, since they had shown a strong decrease in their frequency in the two selection cycles without protein.

The bioinformatics analysis revealed that many aptamers were enriched to ovarian cancer protein HE4 with clear enrichment in positive selection to target HE4 compared to counter selection to sample matrix. The most enriched cluster families were enriched from 0.1–2%. (Figure 4). Moreover, three sequences, internally named AHE1, AHE2 and AHE3 had a clear systematic enrichment to target HE4 (Figure 4) starting from C5+ to the last cycle of selection (with frequency 2.7% for AHE1, 1.4% for AHE2 and 1.2% for AHE3 in C9+ and 1.5% for AHE1, 0.7% for AHE2 and 0.9% for AHE3 in C10+, respectively). In addition, the same aptamers were not being enriched in the counter selection (with frequency <0.4% in C9− and C10− for all three aptamers), indicating them as the most promising binders to human HE4. For this purpose, we have synthesized the most abundant sequences of these three families, internally named AHE1, AHE2 and AHE3.

Based on the enrichment to target HE4, the differences between enrichment in positive versus negative selection (Figure 3 and Figure 4) and secondary structures that could have impact on the binding (Figure 5), we selected candidates AHE1, AHE2 and AHE3 for characterization of the binding affinity to HE4 in urine. Preliminary data showed AHE1 and AHE3 as more potent binders, while AHE2 did not exhibit binding. While it was disappointed to see that AHE2 did not exhibit binding in TFA analysis, its higher GC content (60% in AHE2 vs. 53 and 50% for AHE1 and AHE3, respectively), as well as its C-nucleotide repartition may have introduced a favorable bias in PCR amplification. This C-rich patch may also have provided a sticky hand for interacting with G-rich DNAs during the selection process without direct interaction with the target HE4, which make sequence easily amplifiable, but not necessarily most specific to the HE4 protein target. However, more characterization experiments with optimized conditions (denaturation, Mg concentration or urine dilution) would be needed in future to assess the binding to HE4. For these reasons, two candidate sequences, AHE1 and AHE3, were further characterized.

### 3.3. Thermofluorimetric Analysis of the Aptamers-HE4 Binding

As studies have shown, elevated levels of the protein biomarker HE4 were found in urine of ovarian cancer patients and urine can serve as a non-invasive fluid to detect HE4. Therefore, we wanted to determine the binding constant of anti-HE4 aptamers in urine. For this purpose, artificial human urine containing a constant 100 nM aptamer was spiked with increasing HE4 with a concentration ranging from 0 to 800 nM. To reduce urine interference, while securing the optimal fluorescent signal, urine dilutions were tested. The results showed the interference of 1× concentrated urine, yielding high background and lower signal while showing no effect in 1/125 diluted urine (Appendix A). The Sybr-Gold was binding to DNA aptamers with fluorescence signal directly proportional to aptamer quantity (Appendix A), while no signal was observed with HE4 protein only (Appendix A). These data confirmed the specificity of the signal corresponding to the aptamers-HE4 complex. TFA can thus be used for determination of the binding constants of anti-HE4 aptamers to HE4 in urine. The scramble DNA aptamer sequence was used as a negative control to assess binding specificity.

The negative derivative data −d(RFU)/dT showed that a more thermally stable aptamer-HE4 complex was formed in urine upon binding to target ovarian cancer biomarker HE4 (Figure 6A), as two peaks, corresponding to the free aptamer state and the bound aptamer state, were detected. Appearance of the aptamer-HE4 complex peak was visible around 55 °C for AHE1 and 61 °C for AHE3, versus the peak of free, unbound aptamers at T_m_ (free AHE1) = 47.4 ± 2.6 °C and T_m_ (free AHE3) = 59.9 ± 0.6 °C, respectively. The biggest difference in melting temperature upon binding to HE4 was observed for AHE1 sequence with increase in stability and ΔT_m_ of +7.5 °C, suggesting significant changes in tertiary structure of the aptamer AHE1 upon binding. AHE1 exhibited binding to HE4 protein at a concentration as low as 50 nM of HE4, which corresponded to aptamer-protein ratio 1:0.5. For the AHE3 sequence, the shift of the ΔT_m_ of +4 °C was observed and binding appeared to be significant at higher concentration of HE4, with the AHE3-HE4 complex Tm peak identified at a concentration of 100 nM HE4, which corresponded to a aptamer-protein ratio of 1:1. Subtraction of blank (aptamer only) (Figure 6B) facilitated the observation of a clear shift upon introducing HE4 protein in urine with the new melting temperature of bound aptamer T_m_ (bound AHE1) = of 60 °C and T_m_ (bound AHE3) = of 63 °C. The binding curve was constructed by plotting the concentration of HE4 protein versus the negative derivative fluorescence −d(RFU)/dT signal at the aptamer-protein complex T_m_, which corresponded to 60 °C for aptamer AHE1 and 63 °C for aptamer AHE3, respectively (Figure 6).

For both aptamers, the determined K_d_ is in the nanomolar range in diluted urine, showing they could be potential candidates for urine diagnostic tests. For AHE1, the determined K_d_ (AHE1) was 87 ± 9 nM, with the saturation clearly visible at a concentration of HE4 of 200 nM; while for AHE3, the K_d_ (AHE3) was 127 ± 28 nM, and with the saturation of HE4 binding to AHE3 was observed starting from 300 nM. The negative control, scrambled aptamer DNA, did not exhibit significant changes in T_m_ upon introducing HE4 protein, suggesting no affinity to HE4 and a sequence specific binding of AHE1 and AHE3 to HE4. Plotting of the scrambled aptamer signal against HE4 concentration showed no significant binding, therefore suggesting specificity of binding of AHE1 and AHE3 aptamers to their target HE4 (Figure 7).

## 4. Discussion

Ovarian cancer is accounting for a high number of cancer-related deaths worldwide. One of the main barriers in improving the prognosis is the lack of screening or accurate diagnostic methods. Moreover, currently available diagnostic methods lack the specificity and sensitivity to detect OC in the early stages of this malignant disease. Therefore, novel diagnostic approaches are needed. HE4 is an important clinical biomarker of ovarian cancer. Serum HE4 is routinely employed in differential diagnoses of women with pelvic masses and its clinical utility has been validated [11]. Recently, urine emerged as an easy, non-invasive alternative source of HE4 protein [22]. Urine tests could present easy, cheap and non-invasive alternatives in OC diagnostics. There is no homeostasis mechanism in urine, so excreted cancer biomarker levels stay stable, often intact, and therefore reliably reflect the in vivo pathophysiologic state [18]. Urine can accommodate more changes in the early stage of disease, so urinary analysis could possibly lead to earlier cancer diagnosis, improvement of patient care and reduce a death toll. Urine tests are easy to perform and also have the potential to be used in a self-sampling manner or home setting, opening the door for potential screening strategies on asymptomatic women. The current methods for HE4 are antibody-based immunoassays. The potential limitation of urine tests, including in detection of HE4 protein, is the standardization of the urine concentration. The concentration of HE4 protein will depend on urine volume, which is highly heterogeneous between patients. This is observed by using currently available immunoassays. These conventional tests sometimes require application of creatinine as an internal standard for volume normalization [20,38]. The aptamers combine many advantages, they are low-molecular-weight substances, with high affinity, specificity and production at low cost. For all these reasons, it seems important to explore the use of aptamers for disease diagnosis as a tool that could be validated and standardized in clinical use in future. Aptamers are promising tools for cancer detection in biological fluids, as they can potentially be selected to recognize any oncological biomarker or molecule at low target concentrations which are present in cancer patients [25]. Due to their high affinity to specific targets and high versatility in diagnostic applications, aptamers offer advantages over antibodies in diagnostics. They are extremely stable, which is an important aspect in the urine environment. The nucleic acid structure possesses high stability at the different temperatures in transport and storage, which is convenient for diagnostic tests. On the other side, antibodies are sensitive to temperature changes. Moreover, compared to antibodies, they are easily chemically synthesized and do not require the use of animals. Therefore, low-cost chemical synthesis enssures uniformity in batch-to-batch production. Moreover, aptamers have adaptability and unlimited options of chemical modifications, paving the way to various detection systems in the development of future urine tests and biosensors. Previously, aptamers have been applied in the urine environment in detection of pharmaceutical compounds, such as antibiotics and opioids. Recently, a sensitive DNA aptasensor for detection of oncomarker in urine was developed for colorectal cancer [39].

Herein, aptamers are shown to be functional and stable in the urine environment and exhibit binding to a urinary cancer biomarker target. Up to date, no aptamer has been developed or applied for detection of ovarian cancer in urine. We have demonstrated novel aptamer probes that function in urine, with high-affinity binding to ovarian cancer biomarker HE4, holding the potential to application in future diagnostic tests. The aptamers were developed after 10 rounds of selection to HE4 in 1× urine. As the final goal is to apply these aptamers to function as diagnostic probes in human urine, it was important to ensure a selection environment of high salts and slightly acidic pH equivalent to urine, which will provide the environment for the relevant three-dimensional structures of aptamers binding to HE4.

The aptamers were successfully developed and identified using ddPCR-based Hi-Fi SELEX. We used a modified version of a previously published method [30] and expanded to droplet analysis and possibility of ddPCR aptamer quantification. As library members are partitioned into droplets and each droplet represents individual PCR reaction, ddPCR ensures sensitive amplification of very rare sequences, minimizing the loss of potentially good and specific binders to target cancer biomarkers. Additionally, it can offer absolute quantification of aptamers during selection, without the need for standards. It eliminates amplification artifacts, often present as an issue in SELEX. Using ddPCR, each library member exists in a complementary duplex, ensuring proper ssDNA regeneration at the end of the cycle. Therefore, potentially less cycles are needed for a selection of high-affinity binders [30]. The main limitations observed are the high cost and low recovery of DNA. As ddPCR was not originally intended for downstream analysis (once droplets are analyzed, they are discarded), re-amplification is needed to increase the yield of DNA to be able to proceed in the next cycle of SELEX. In this case, ddPCR ensures sensitive and non-biased pre-amplification before classical PCR. Since ddPCR is an extremely sensitive method enabled to amplify only a few aptamer sequences present in sample (i.e., 3 present bound sequences in 20 µL = 0.15 copy/µL will result in 3 positive droplets), it is important to achieve optimal template concentration to maximize the effect of partitioning capabilities and recovering of all relevant specific aptamer library members. Therefore, we would recommend the dilution of bound aptamer fractions and separation in multiple reactions rather than adding the total quantity of a bound DNA template. The presence of a high amount of template will inhibit the PCR efficiency and saturate the droplets, as to what is observed in the first few cycles of selection, and reduce the capacity to fully amplify rare sequences. Therefore, decreasing the aptamer template is recommended at the beginning of SELEX cycles using ddPCR. ddPCR can quantify the anti-HE4 sequences, but it does not provide information in which sequences are present. For that reason, DNA sequencing is always crucial to be able to identify specific enrichment to target cancer biomarkers. Hi-Fi SELEX was successful in selection and identification of high-affinity aptamers to ovarian cancer biomarker HE4. Our findings provide proof-of-concept for using ddPCR and urine for aptamer selection and can be applied for future development of any novel diagnostic or therapeutic aptamers.

After selection, the two most promising candidates, called AHE1 and AHE3, were tested for binding in urine. As elevated concentrations of HE4 are present in patients, it is important to validate binding of anti-HE4 aptamer probes in urine. Artificial human urine was spiked with HE4 protein, and the aptamers AHE1 and AHE3 showed high affinity to target HE4 in urine, with K_d_ in the nanomolar range. The results further proved the concept of utilization of aptamers as diagnostic probes in urine for the development of future OC urine tests. The mean urine levels of urine HE4 present in patients with ovarian cancer described in literature are 28.56 nM [21] and 29.83 nM [38]. The cut-off value which is able to distinguish healthy individuals from cancer patients is approximately 13 nM [23], so binding of aptamers in urine should ideally be observed at concentrations >13 nM urinary HE4, which correspond to cancer patients. Indeed, the K_d_ value of aptamers AHE1 of 87 ± 9 nM Kd and aptamer AHE3 of 127 ± 3 nM in urine shows binding in nanomolar range, on a scale relevant to HE4 concentrations in clinical samples of ovarian cancer. Although presented data suggested the potential of described aptamers, more studies are needed to validate the specificity of described aptamers, including oligomerization and optimization of the sequence to ensure detection within a clinical range. Further characterization and optimization of the sequences is ongoing. Future developments include testing with actual human patient samples and at very low concentrations on picomolar range. The described DNA aptamers with high binding affinity to urine HE4, reflected by nanomolar K_d_ values, will be developed for the future use in aptamer-based urine bioassays and biosensors for ovarian cancer.

## 5. Conclusions

Only about 20% of ovarian cancers are found at an early stage. Finding ovarian cancer early would improve poor prognosis and reduce the high mortality. Therefore, we want to explore the possibility for early detection using aptamers as probes targeting urinary human epididymis protein 4. HE4 protein is a specific biomarker which is overexpressed in ovarian cancer and present in urine.

Here, we selected new DNA aptamers, named AHE1 and AHE3, binding with high affinity to human HE4 protein in urine. The anti-HE4 aptamers were selected using Hi-Fi SELEX, a selection method based on digital droplet PCR partitioning and sensitive amplification of rare aptamer sequences. The method allowed to isolate the aptamers able to effectively target HE4 protein in the urine environment.

As HE4 levels are elevated in the urine of patients, the aptamers were further characterized in urine and exhibited good affinity to target HE4, with a constant of dissociation in the nanomolar range. These results suggested that AHE1 and AHE3 are promising diagnostic probes that can be further used in the development of diagnostic tests and biosensors for the detection of ovarian cancer.

## Figures and Tables

**Figure 1 cancers-15-00452-f001:**
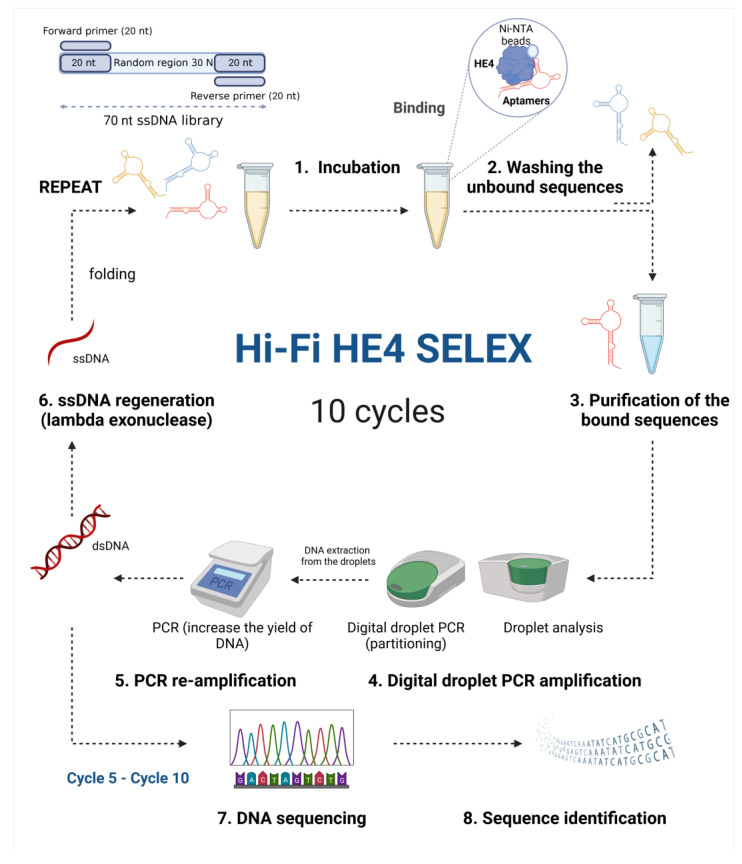
Schematic representation of the digital droplet PCR (ddPCR)-based High Fidelity Systematic Evolution of Ligands by EXponential enrichment (Hi-Fi SELEX) method for the selection of aptamers to ovarian cancer biomarker Human epididymis protein 4 in urine. High-affinity aptamers are selected from initial 70 nucleotide long (70 nt) single-stranded DNA library consisting of random 30 nucleotide long region (30 N), flanked by 20 nucleotide long (20 nt) constant regions used for PCR amplification. Briefly, the aptamer libraries are incubated with target HE4 protein in urine. Non-specific unbound sequences are washed, while specific bound sequences are recovered and purified. The specific anti-HE4 sequences are partitioned into droplets for sensitive ddPCR amplification. Then, the aptamers are recovered from the droplets and reamplified using regular PCR to increase the yield of DNA. Double-stranded PCR products are digested by lambda exouclease to the single-stranded sequences used as an input library for the next cycle of selection (Created with Biorender.com).

**Figure 2 cancers-15-00452-f002:**
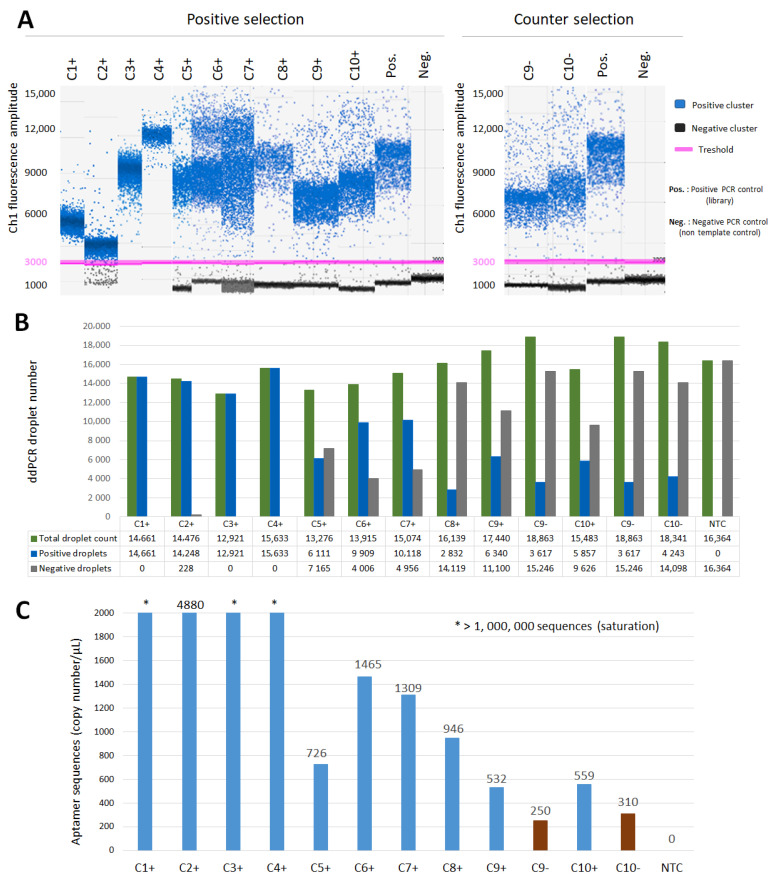
Digital droplet PCR analysis of aptamers recovered during Hi-Fi SELEX to ovarian cancer biomarker HE4. (**A**) 1-D plot of Hi-Fi SELEX positive selection to target HE4 protein and counter selection to 6xhistidine peptide. Many sequences were detected in all cycles of selection. In the plot, each droplet from a sample in each SELEX cycle is plotted on the graph of the fluorescence intensity vs. droplet number. All positive droplets with anti-HE4 aptamer present (in blue), are those above the pink threshold line and are scored as positive. All negative droplets are those without target aptamers present (in dark gray) below the red threshold line and scored as negative. (**B**) ddPCR droplet enumeration throughout Hi-Fi SELEX positive and negative selection. The droplets were classified and quantified as positive (target anti-HE4 aptamers present), negative (no target aptamers) and total (positive + negative droplets). In the first few selection cycles, all droplets were positive (saturation) due to the excess of aptamer sequences present. After cycle 5, repartition was optimal, with positive and negative droplets present, due to the decrease of aptamers present. Decrease of aptamer sequences present equaled increase of sequence specificity to HE4 protein. As SELEX evolved, many specific sequences were being enriched to target HE4 protein. (**C**) Quantification of the anti-HE4 aptamers throughout Hi-Fi SELEX positive and negative selection. Numerous sequences were detected and quantified in each cycle. At the beginning of selection, a maximum number of sequences were observed (saturation). After cycle 5, sequence number and diversity decreased and specificity to HE4 increased. Most importantly, a higher number of sequences were observed as enriched in positive selection to target 6xhistidine-HE4 protein compared to counter-selection to 6xhistidine peptide (sample matrix) in both cycle 9 and 10, suggesting that enrichment of specific sequences to HE4 occurred.

**Figure 3 cancers-15-00452-f003:**
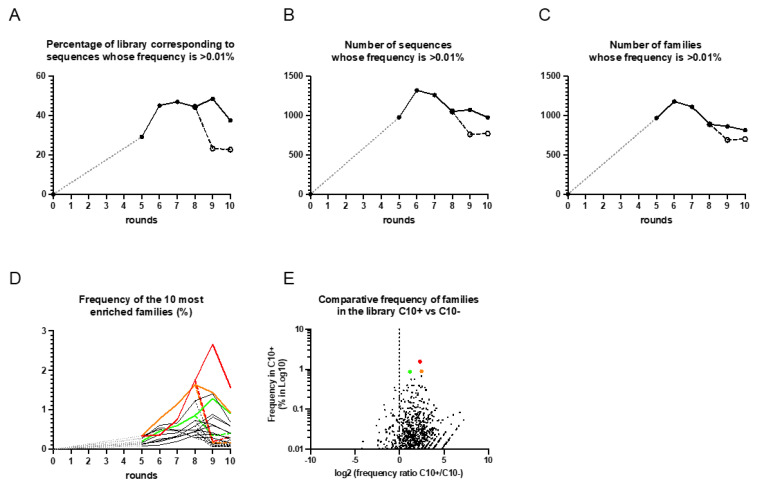
Deep sequencing analysis of SELEX. The starting library and the libraries after cycle 5 were analyzed. Sequences whose frequency is higher than 0.01% at least in one cycle were recovered and their percentage in the library as well as their number were measured for several cycles (**A**,**B**), respectively). Those sequences have been clustered in families based on an edit distance of 6. The evolution of the number of families is presented in (**C**). The frequency of the 10 most enriched families is presented in (**D**). The frequency of each family in the last C1+ cycle relative to its enrichment in the presence or absence of the HE4 protein (C10+/C10− ratio) is presented in (**E**). The evolution in solid lines and black dots correspond to libraries from positive selection in the presence of HE4 protein, while the evolution of dashed lines and light dots correspond to the two cycles without protein. The gray dotted lines correspond to cycles that were not sequenced. The colored curves and dots correspond to the evolution of the 3 families which were selected for binding evaluation (red, orange and green for AHE1, AHE2 and AHE3, respectively).

**Figure 4 cancers-15-00452-f004:**
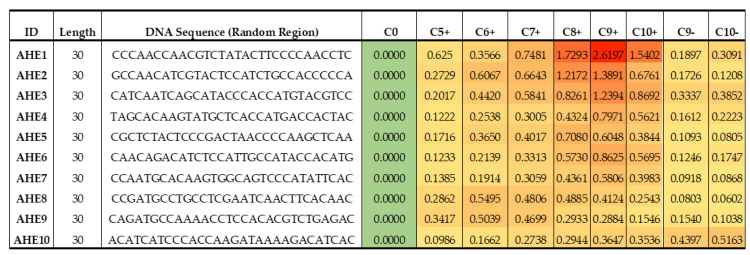
Evolution of the top 10 most enriched DNA aptamers targeting ovarian cancer biomarker HE4 obtained with DNA sequencing. The results show clear enrichment of certain families (especially AHE1, AHE2 and AHE3) after positive selection to target HE4 protein in urine (from cycles C5+ to C10+), but not in counter-selection (C9− and C10−). C0 represents native library, C5+ to C10+ represent positive selection to target protein 6xhistidine-HE4 protein and C9− and C10− represent counter-selection to 6xhistidine peptide and beads (sample matrix). Green color = no enrichment to HE4, yellow = intermediate enrichment, orange = mild enrichment, red = intense enrichment to HE4 protein.

**Figure 5 cancers-15-00452-f005:**
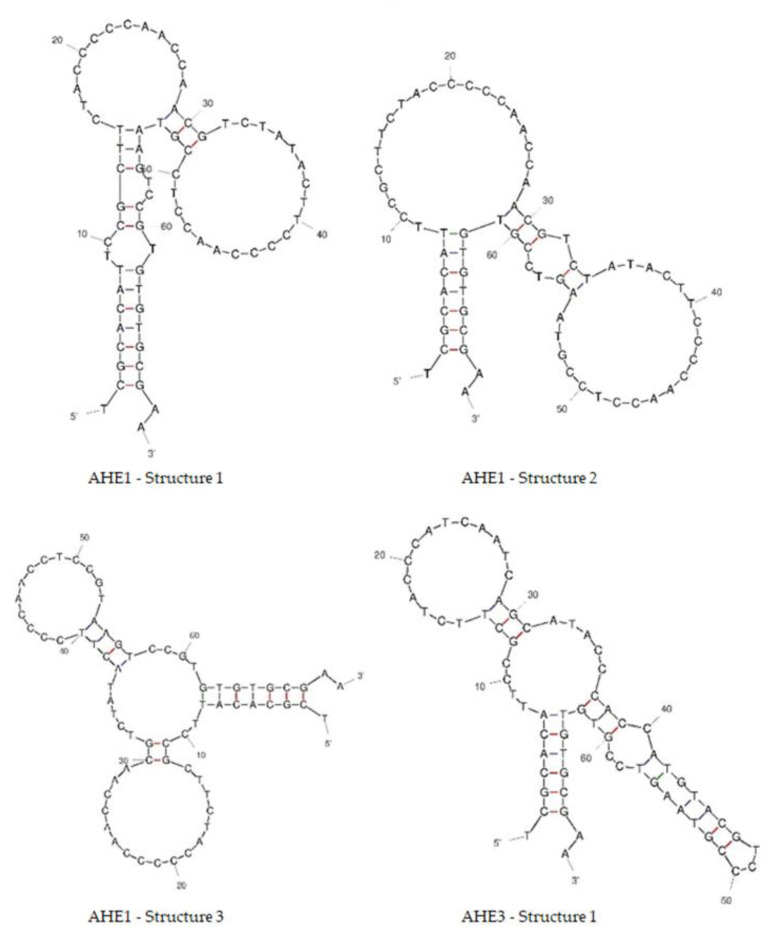
Predicted secondary structures of anti-HE4 aptamers. The secondary structures of DNA aptamers (AHE1 have three potential structures, AHE3 has one potential structure predicted) including full length 70-mer sequences were created using online DNA folding software Unafold (http://www.unafold.org) at temperature 25 °C and urine concentration of [Na^+^] = 55.4 mM and [Mg^2+^] = 4.4 mM.

**Figure 6 cancers-15-00452-f006:**
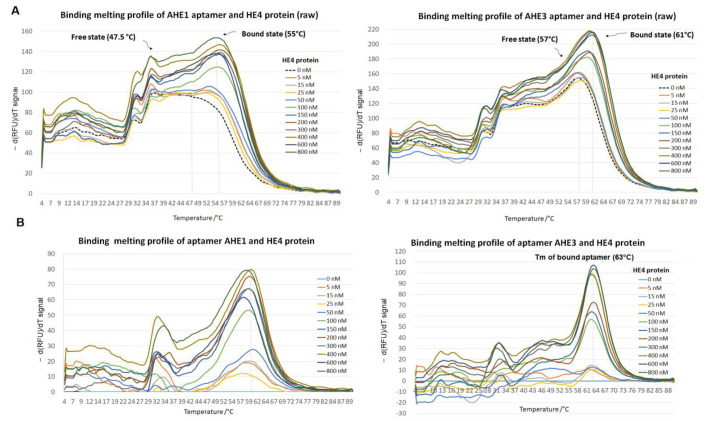
Thermofluorimetic analysis of the anti-HE4 aptamer binding to ovarian cancer biomarker HE4 in urine. The constant aptamer concentration of 100 nM was subjected to an increasing concentration of HE4 protein in urine ranging from 0 to 800 nM. The melting profile was analyzed from 4 °C to 90 °C. The binding thermal curves (melting DNA profile) were constructed by plotting temperature with negative derivative fluorescent signal −d(RFU)/dT. The results showed the average of the minimum 3 independent experiments. (**A**) The raw melting profile. The raw melting profile obtained for aptamers showed two distant peaks corresponding to free aptamer (no HE4) state and bound aptamer state (with HE4 protein). Upon binding to HE4 protein, more thermally stable species were present, with a shift to higher T_m_. (**B**) The melting profile after subtraction of signal from aptamer only. After subtraction of the blank (aptamer only, no HE4), peaks of the aptamers bound to HE4 were visible, with a shift to higher T_m_ values, corresponding to 60 °C for AHE1 and 63 °C for AHE3, suggesting binding to target protein HE4.

**Figure 7 cancers-15-00452-f007:**
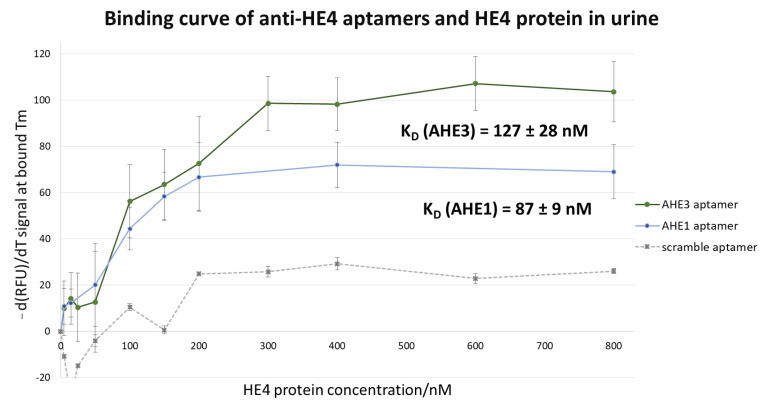
Binding curve of anti-HE4 aptamers to ovarian cancer biomarker HE4 in 1/125 urine. The binding curve was constructed by plotting HE4 protein concentration with negative derivative fluorescence signal −d(RFU)/dT at the T_m_ corresponding to bound aptamer state. The non-linear regression model was used to calculate K_d_ value, showing good affinity in nanomolar range of anti-HE4 aptamers for HE4 in urine, with K_d_ (AHE1) = 87 ± 9 nM and K_d_ (AHE3) aptamer of 127 ± 28 nM. The results showed the average of the minimum 3 independent experiments. The results suggest that the described aptamers could have diagnostic potential as detection probes for HE4 in urine tests for ovarian cancer.

**Table 1 cancers-15-00452-t001:** Overview of the Hi-Fi SELEX conditions for selection of aptamers to ovarian cancer protein biomarker HE4 in urine. The aptamers were selected after 10 cycles of positive selection (+) to target 6xhistidine-HE4 and two rounds of counter selection (−) to 6xhistidine peptide. To obtain specific sequences, the selection stringency is increased by increasing the amount of DNA and decreasing the amount of protein.

Cycle	DNA Source	DNA (nmol)	Protein (pmol)	Ratio
C1+	library	1.25	200	6.25:1
C2+	C1+	0.47	200	2.35:1
C3+	C2+	0.08	100	0.80:1
C4+	C3+	0.13	100	1.30:1
C5+	C4+	0.15	100	1.50:1
C6+	C5+	0.04	200	0.20:1
C7+	C6+	0.03	200	0.15:1
C8+	C7+	0.08	200	0.40:1
C9+	C8+	0.10	200	0.50:1
C9−	C8+	0.10	200	0.50:1
C10+	C9+	0.10	100	1.00:1
C10−	C9−	0.10	100	1.00:1

## Data Availability

The data that support the findings of this study are available from the corresponding author upon reasonable request.

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
