# Peer review of "Identification and Characterization of Aptamers Targeting Ovarian Cancer Biomarker Human Epididymis Protein 4 for the Application in Urine"

_cancers, 2023, doi:10.3390/cancers15020452_

Round 1
Reviewer 1 Report
I have read with interest the manuscript by Antonija Hanžek et al. entitled “Identification and characterization of aptamers targeting ovarian cancer biomarker Human epididymis protein 4 for the application in urine”. While I appreciate the intent of the work there are several concerns and suggestions that the authors should accept in order to prepare the MS for publications.
1) Statement about diagnostic methods of ovarian cancer is not actual (lines 48-50). Currently, golden standard in OC diagnostic procedures are gynecological examinations and blood test using ROMA and/or He4 measurement. Please check the adequate information with new recommendations of gynecological associations.
2) Additional information about the volumes of PCR mixtures (in the tab.1 in manuscript or supplementary materials) should be added.
3) Please, check the amounts of libraries used in each round of aptamer selection. The amounts provided by the authors seems to be to high.
4) The title of MS suggests potential use of application of aptamers for detection of He4 in human urine. Still the presented results are not related to urine samples measurements. The authors analyze buffer with the composition similar to human urine. Thus, additional experiments that demonstrate possibility of He4 measurement in urine samples should be performed.
Reviewer 2 Report
In the present paper, the authors identify and characterize aptamers that recognize the HE4 protein for its potential use in diagnostic urine samples that would be of great relevance in ovarian cancer.
The work is well written, with clear background, and of high relevance. I have 2 important comments for the methodology and some less relevant structure and wording.
1. The information presented between lines 390-393 is not sufficiently precise. The search for motives by means of multiple alignments does not have enough robustness as other software for the identification of specific motives (MEME suite and MOTIF finder for example)
2. It is not surprising that one of the candidates with the highest representativeness within the libraries, AHE2, does not demonstrate any binding by the HE4 protein. I think we need a more detailed explanation of what could have happened with this. If it is a detected failure of the SELEX process or operator error or other external problem that does not compromise the quality of the SELEX experiment performed.
Minor comments:
>Correct cursive error (line 84) in vitro
>In figure 1 differentiate between nt (nucleotides) and random sequences (N) and make it clear in the figure caption
>Unifying units of measurement in some places is used uL (line 193) and in others ul (line 215) and in many other places in the full text
>The size of the variable region would not have to be modified along the SELEX, because the region is considered to be between 25-32 nucleotides on line 252
>In line 438 there is a missing number for AHE, apparently the authors refer to AHE1
Round 2
Reviewer 1 Report
Dear Authors,
Thank you for the improvements introduced in the manuscript.
Tab 1. Amounts of DNA are incorrect, please recalculate the amounts of DNA.
(According to the information for C5 cycle amount of DNA was 151 nmol, what corresponds to 3,17 mg of DNA)
(Average DNA molecular weight 70 x 300 =21000)
Supplementary data
Table S1. A B.
Detailed constitution and volumes of PCR mixtures must be improved.
Units of primer concentration (10 µM) /µL, what does it mean?
